# An Urban Autodriving Algorithm Based on a Sensor-Weighted Integration Field with Deep Learning

**Minho Oh**, **Bokyung Cha, Inhwan Bae, Gyeungho Choi * and Yongseob Lim ***

Department of Robotics engineering, Daegu Gyeongbuk Institute of Science & Technology (DGIST),
Daegu 42988, Korea; dhalsgh5@dgist.ac.kr (M.O.); lcbk_0321@dgist.ac.kr (B.C.); bih9907@dgist.ac.kr (I.B.)
* Correspondence: ghchoi@dgist.ac.kr (G.C.); yslim73@dgist.ac.kr (Y.L.)

**Abstract:** This paper proposes two algorithms for adaptive driving in urban environments: the first uses vision deep learning, which is named the sparse spatial convolutional neural network (SSCNN); and the second uses a sensor integration algorithm, named the sensor-weighted integration field (SWIF). These algorithms utilize three kinds of sensors, namely vision, Light Detection and Range (LiDAR), and GPS sensors, and decide critical motions for autonomous vehicle, such as steering angles and vehicle speed. SSCNN, which is used for lane recognition, has 2.7 times faster processing speed than the existing spatial CNN method. Additionally, the dataset for SSCNN was constructed by considering both normal and abnormal driving in 7 classes. Thus, lanes can be recognized by extending lanes for special characteristics in urban settings, in which the lanes can be obscured or erased, or the vehicle can drive in any direction. SWIF generates a two-dimensional matrix, in which elements are weighted by integrating both the object data from LiDAR and waypoints from GPS based on detected lanes. These weights are the integers, indicating the degree of safety. Based on the field formed by SWIF, the safe trajectories for two vehicles' motions, steering angles, and vehicle speed are generated by applying the cost field. Additionally, to flexibly follow the desired steering angle and vehicle speed, the Proportional-Integral-Differential (PID) control is moderated by an integral anti-windup scheme. Consequently, as the dataset considers characteristics of the urban environment, SSCNN is able to be adopted for lane recognition on urban roads. The SWIF algorithm is also useful for flexible driving owing to the high efficiency of its sensor integration, including having a resolution of 2 cm per pixel and speed of 24 fps. Thus, a vehicle can be successfully maneuvered with minimized steering angle change, without lane or route departure, and without obstacle collision in the presence of diverse disturbances in urban road conditions.

**Keywords:** autonomous driving; sensor integration; SWIF; vision deep learning; SSCNN; lane recognition; obstacle detection; maneuvering control

---

## 1. Introduction

In the last three decades, studies on autonomous driving have made remarkable progress due to the efforts of many researchers [1,2]. In the field, driver convenience is enhanced by providing adaptive cruise control to maintain a constant vehicle speed and a highway driving assist system to prevent lane departure on highways [3]. However, research studies on urban environments are still scarce compared to highways, which are relatively simple environments mostly containing vehicles and roads. Unlike highways, various dangerous situations happen in urban environments. Obstacles such as construction sites often block roads or cover lanes, as shown in Figure 1a, and sometimes pedestrians appear suddenly on the road (Figure 1b). In addition, vehicles close to the opposite lane

can be critically dangerous to other vehicles in the adjacent lane. Thus, autonomous driving algorithms must guarantee safety by making flexible decisions [4,5].

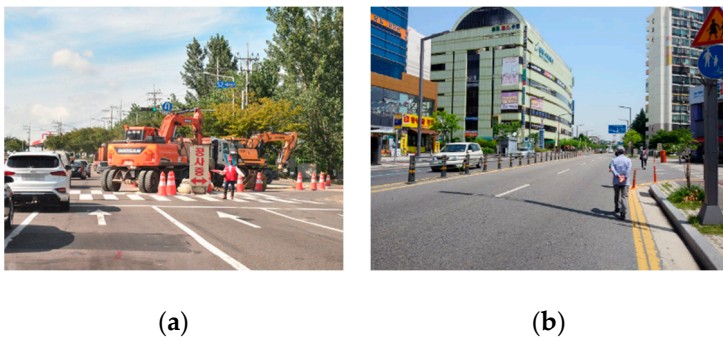

      (**a**)                           (**b**)

**Figure 1.** Disturbance factors on a road: (**a**) construction site; (**b**) pedestrian on the road.

One of the difficult tasks in autonomous driving is lane recognition when considering extreme situations, such as crowded roads, shadows, and dazzling light. [6]. Until recently, most conventional algorithms required mathematical description of low-level features, such as color patterns and straight lines [7,8]. Adopting deep learning has led to extraction of lanes without feature description. Among the vision deep learning methods, recurrent neural networks (RNN) pass information along each row or column so that each pixel can only receive information from the same row or column [9,10]. Zhang et al. proposed this design in order to learn the geometrical conditions that make up a road, including lane boundaries and road areas. This design performs lane boundary segmentation and road area segmentation at the same time [11]. On the other hand, spatial convolutional neural networks (SCNNs) implement sequential message passing to utilize structural information [6]. Although parts of traffic lanes in urban environments could be obscured by obstacles (e.g., other vehicles, traffic cones, shadows, etc.), this method predicts the obscured lane locations by extending the other parts of traffic lanes that are not obscured. Moreover, performance of SCNN in terms of processing time can be improved by changing the SCNN network model to be sparser.

To determine the motion of autonomous vehicles, information about road conditions is integrated. Search- and sampling-based planning and optimization are the ways to determine a path by integrating information from sensors [12–14]. These path planning algorithms divide the space into compartments and traverses, checking the state identified by the sensor information. This state space is usually represented as an occupancy grid or lattice [15]. The graph searching algorithm gives the solution based on the evaluation criteria. The A* algorithm, one of the search-based algorithms, decides on the path by using the Voronoi cost function to define the path length and proximity to obstacles [16,17]. This algorithm works well for searching paths in a known space. However, it has poor memory efficiency and performance in large areas [12].

Recently, technologies connecting other facilities around autonomous vehicles have been developed. Especially, vehicle-to-everything (V2X) communication is an essential technique in intelligent transport systems (ITS) to provide real-time information (e.g., traffic conditions and accidents [18–21]). For example, by utilizing the compensated route waypoints and V2X technologies, critical performance (relating to road safety, driving efficiency, situational awareness, etc.) is enhanced in highly demanding situations, such as at intersections and during lane changes.

In this paper, a flexible path planning algorithm is proposed for urban environments, using a novel sensor integration method, namely the sensor-weighted integration field (SWIF) method, which is based on the newly proposed vision deep learning technique (i.e., the spare spatial convolution neural network (SSCNN) technique). By processing all sensor data, including vision, LiDAR, and GPS data, the SWIF algorithm provides a 2D field measuring 300 pixels × 300 pixels (6m × 6m in distance units). The proposed algorithm is capable of assigning different weights to each pixel, with higher weights representing safer areas. In particular, utilizing SSCNN enables not only highly accurate

lane recognition, but also means that obscured or erased traffic lanes can be detected based on the recognized lane parts. In addition, traffic lanes are also recognized both in normal maneuvering and in abnormal maneuvering scenarios. Moreover, both SWIF and SSCNN are developed to become real-time systems by making the network structures and functions sparse. Based on the 2D weighted field method and as a result of SWIF and SSCNN, a cost field is implemented to derive the minimal steering angle needed to avoid obstacles without route or lane departure. Thus, this algorithm enables safe and flexible motion planning in real time, and was verified through experiments in urban scenarios. Consequently, the proposed algorithm enables safe driving and allows for sudden changes.

The paper is organized as follows. In Section 2, the specifications of sensors used in the experiments are introduced. In Section 3.1, after reviewing existing the SCNN model, the SSCNN network model for fast lane recognition is explained. In Section 3.2, the SWIF algorithm is introduced. In Section 3.3, the motion planning and control method used to generate steering angles and vehicle speeds is described. In Section 4, experimental results are demonstrated. In Section 5, conclusions are described.

## 2. Sensor Setup

Three types of sensors were used in this algorithm. First, a Logitech c930e sensor was used as a vision sensor for lane recognition. It was located in the highest position to recognize a minimum of three traffic lanes. Second, a 2D-LiDAR sensor, which is used for object detection, was located at the front and at the lowest point possible of the vehicle. For this reason, the blind spot at the front could be minimized. Third, the GPS sensor, located at the top of the vehicle, was used for both localization on the map and also used for setting the waypoints. These three sensors were all located in the vertical center of the vehicle, and their specifications are shown in Table 1. Moreover, the road coverage by the sensors is shown in Figure 2.

**Table 1.** Specifications of sensors.

| Sensor | Specification |
|---|---|
| Vision (Logitech c930e) | Viewing angle of 90 degrees Resolution of 1080p with 30 fps |
| 2D LiDAR (LMS 151) | Recognition distance up to 50 m Resolution of 0.25 degrees Scanning speed of 50 Hz |
| Real Time Kinematic GPS (MRP-2000) | Position Accuracy: 0.01 m in horizontal; 0.01 m in vertical Time to First Fix: 28 s (in Digital Multimedia Broadcasting mode) |

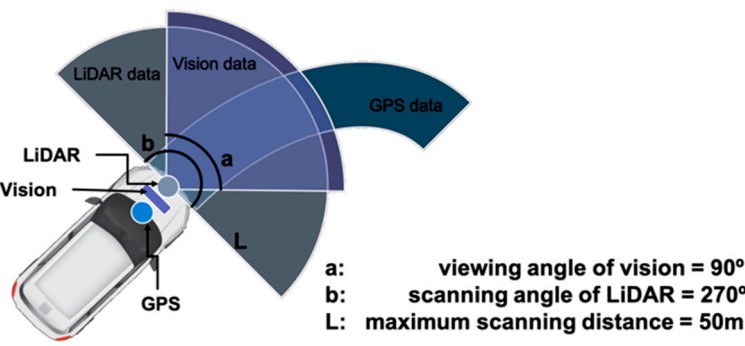

**Figure 2.** Sensor arrangement and coverage.

## 3. Proposed Methodology

Three types of sensor data and map information were integrated for the recognition of diverse road environments. The sparse spatial convolutional neural network (SSCNN), a vision deep learning method, was proposed for lane recognition. It is capable of detecting the adjacent four traffic lines

in any driving direction. The route from origin to destination in an absolute coordinate is obtained from the map and transformed into relative route data using GPS. In addition, by utilizing detected object data from LiDAR, the dangerous and safe areas are defined. The above pre-processing data are integrated into the SWIF and are then used for motion planning to determine the steering angle and vehicle speed. The proposed algorithm flow is shown in Figure 3.

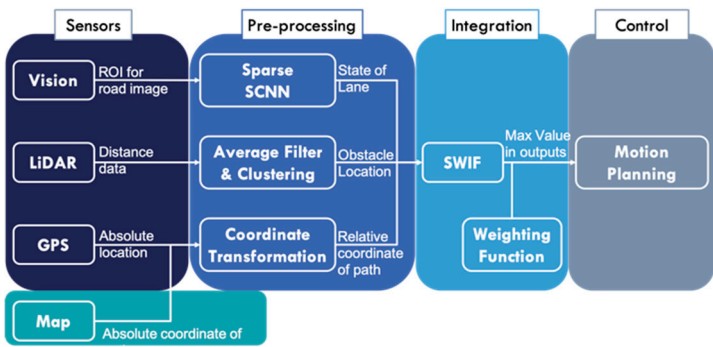

**Figure 3.** Proposed algorithm flow. Note: ROI = region of interest; SWIF = sensor-weighted integration field; SCNN = spatial convolutional neural network.

### 3.1. Vision Deep Learning: Sparse Spatial CNN

On urban roads, there are various types of lane classes, such as straight roads, curved roads, crossroads, and diverse road markings, and the lane markings can be obscured or erased. Moreover, while avoiding or overtaking objects, vehicles may have to maneuver to various locations across diverse roads. Therefore, this section describes the novel dataset, which considers lane extension and maneuvers in many directions, along with SSCNN, which is a lighter and faster deep learning network model than the previous ones. In addition, SSCNN was designed to obtain data for the adjacent four traffic lines from a road image.

#### 3.1.1. Dataset

Existing datasets did not consider steering and movement on the road. Regarding the Caltech lanes dataset (Aly, 2008), Tusimple benchmark dataset (Tusimple, 2017) [22], and the CULane dataset (Pan, 2018) [6] datasets, most of them looked at the vanishing point direction. Moreover, regarding the inclination of a lane due to the steering on the lane, the recognition rate was critically low. Thus, a dataset containing these issues was generated for deep learning.

With the sensor viewing forward and horizontally on the vehicle, the data were collected in the K-City facility, which is built for autonomous vehicle testing in Korea. In total, 7175 frames were extracted from the road image. The urban dataset was separated into 6011 frames as training sets and 264 frames as test sets. The training dataset was also divided into normal and abnormal maneuvering, while classifying the classes into straight roads, curved roads, crossroads, and road markings (e.g., arrows, diamonds, speed bumps, and crosswalks). In general, normal maneuvering means that a vehicle drives properly along the center of the lane, and abnormal maneuvering means that a vehicle staggers in any direction, regardless of lane direction. The images for each classification are shown in Figure 4. As shown in Figure 5, the ratios of each class are represented. Especially, 27.78% of the dataset consisted of abnormal maneuvering, and 72.22% of the dataset consisted of normal maneuvering. Moreover, a total of four adjacent traffic lines were labeled, and road markings were not labeled. The labeling was proceeded by placing points on the coordinates in the image to create a spline through the points.

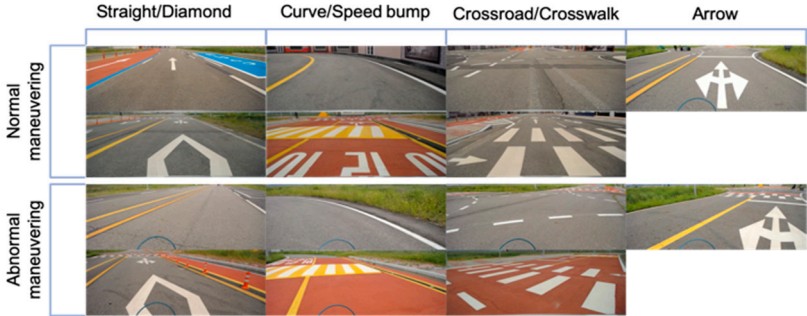

**Figure 4.** Road classification for a sparse spatial convolutional neural network (SSCNN).

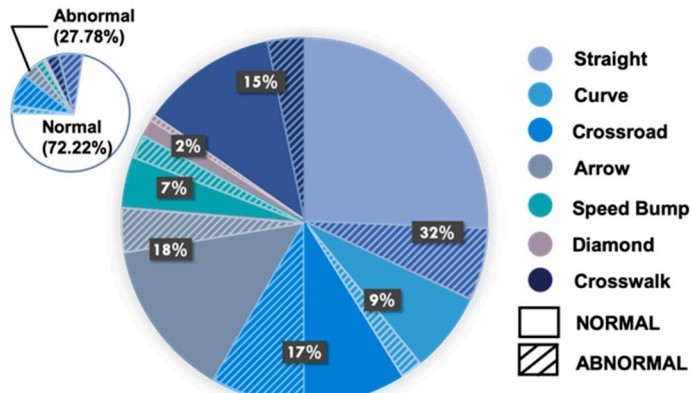

**Figure 5.** Composition ratio of each class of data.

### 3.1.2. Proposed Network Model

Among the most commonly used deep learning methods, the neural network structure of the combined Markov random field–conditional random field (MRF–CRF) method is the basic structure for deep learning. In this structure, information is conveyed between every neuron, as shown in Figure 6a. The structure of the SCNN was designed to transfer information only between neighboring neurons from the next slice, to reduce unnecessary processes and also to reduce execution time. The structure of the SCNN network model is expressed as a 3D tensor with dimensions C × H × W, where C is the number of channels, and H and W are the number of rows and columns, respectively. When the information is conveyed downward or upward in the existing SCNN, the information is transferred immediately to the next row. Moreover, when the information is conveyed rightward or leftward, the information is transferred immediately to the next column (Pan, 2018) [6]. For example, the process of information transfer to the right is simply expressed in Figure 6b. In other words, through SCNN, information is only transferred between neighboring slices.

In order to apply the proposed algorithm to autonomous vehicles in urban environments, it is important to maintain the high performance of the lane recognition rates in all driving environments, including during normal and abnormal maneuvering. In addition, a lighter network model in terms of calculation is required for real-time operation with a high sampling rate. Thus, in order to design a lighter network model in terms of computation, the neural network structure of the spatial convolutional neural network (SCNN), which showed the highest results in lane recognition performance, was modified. When the existing SCNN was used, the computation phase of SCNN took the most time in the entire execution time of the integrated system. Thus, the new network model was proposed to increase the computational efficiency of the existing SCNN. Based on the network structure of the SCNN, the *n* rows or *n* columns are grouped to make the transfer process sparser, and then the information is transferred between the neighboring groups (not slices), as shown in Figure 6d. In other words, when transferring downwards or upwards, the *n* rows are grouped, and information is

conveyed through *H/n* slices. When transmitting rightward or leftward, the *n* columns are grouped, and information is conveyed through *W/n* slices. With this method, the number of transfers was reduced by 1/*n* times. In addition, the structural property of the MRF–CRK, which reprocesses the result and gets a more accurate one, was introduced in each grouped slice, as shown in Figure 6c,e. For example, when information is transferred rightward through the SSCNN from the start to the end of the tensor, the processes shown in Figure 6c–e are performed sequentially. The network model formed in these ways was newly defined as the sparse spatial convolutional neural network (SSCNN). In application of the SSCNN to our proposed system, the number of *n* was set as 2. In Figure 6, $1 \times 4 \times 4$ tensors are represented for simplicity instead of a large-size 3D tensor.

These models were evaluated for urban autonomous driving with the Tusimple benchmark dataset, CULane dataset, and the above dataset. In real-world applications, the predicted center points of each lane were extracted by applying the argmax function to the horizontal slice of the 2D tensor output from the neural network. Additionally, the Gaussian filter was used to smooth the output values. Parameters were manually tuned according to the output tensor size and noise level.

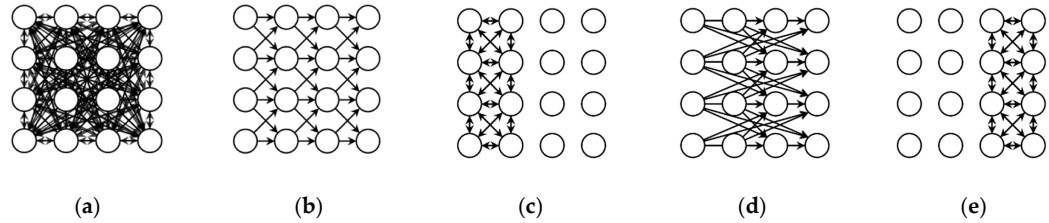

|  (**a**)  |  (**b**)  |  (**c**)  |  (**d**)  |  (**e**)  |

**Figure 6.** Network models: (**a**) Markov random field–conditional random field (MRF–CRF); (**b**) spatial convolutional neural network (SCNN); (**c**–**e**) proposed sparse spatial convolutional neural network (SSCNN).

## 3.2. Proposed Sensor Integration Algorithm: Sensor-Weighted Integration Field (SWIF)

When an autonomous vehicle is traveling, it is important to know not only where the lanes are, but also in which direction to go and which are is safe from dangerous factors. For these factors, a simple and accurate sensor integration method, named SWIF, is proposed, which uses three sensors to recognize the environment, decides where the safer area is in which to maneuver, and then minimizes the overall risk during vehicle travel.

### 3.2.1. Lane Data

The SSCNN identifies the locations of the four adjacent lanes in the road image, which are named "left-left" (or blue line in the image), "left" (or green line in the image), "right" (or red line in the image), and "right-right" (or yellow line in the image), in order from left to right. Moreover, if any of the four lines are not recognized, this part is specified as "none". For example, as shown in Figure 7, only three traffic lines are detected on a two-lane road because there is no "right-right lane". Considering the hardware position of the vision sensor (i.e., viewing angle and height), the detected lanes are changed into curves, as seem from the top by using the warping function in OpenCV. Moreover, the curves are expressed in the field (the size of which is 300 pixels × 300 pixels) by cutting the elements outside the 6 m wide by 6 m long area. This field is defined as a lane field. Moreover, in the lane field, a weighted lane field is formed by applying the weighting factor in the following process shown in Table 2.

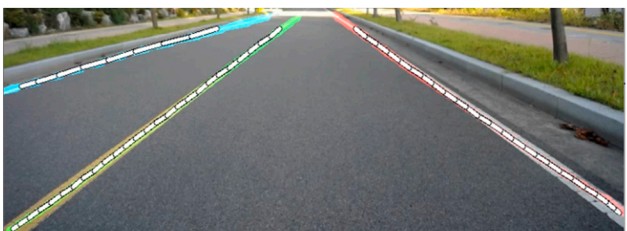

**Figure 7.** Lane detection by SSCNN.

**Table 2.** Pseudocode for forming a weighted lane field.

| | |
|---|---|
| Input: | Lane field |
| Process: | If there is "left" ("right") <br> "left" ("right") is considered as the left lane (or right lane). <br>   "left-left" ("right-right") is ignored. <br> Or else <br> "left-left" ("right-right") is considered as the left lane (or right lane). <br> Only coordinate values of left and right lanes remain in the lane field. <br><br> For each row of the lane field <br> Based on the column component of the left lane, the weights as shown in Figure 8a are assigned. <br> Based on the column component of the right lane, the weights as shown in Figure 8b are assigned. |
| Output: | Weighted lane field |

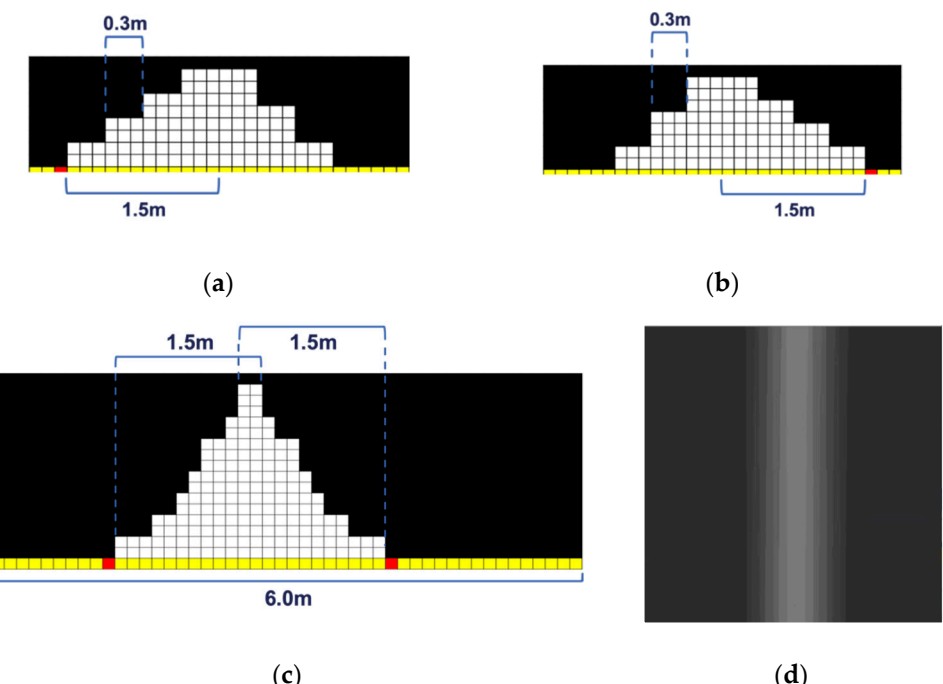

(a)

(b)

(c)

(d)

**Figure 8.** Schematic weighted lane field: (**a**) weights applied to the adjacent left line; (**b**) weights applied to the adjacent right line; (**c**) applied weights in a random row of the weighted lane field; (**d**) weighted lane field from Figure 7.

Figure 8a–c are the schematic diagrams that describe how much weight is applied on each column and each line. Based on the location of the red dot (representing the column component of the left traffic line in a random row), weights are given on the right side of red dot, as shown white area in Figure 8a. Conversely, for the right traffic line, weights are applied on the left side, as shown in

Figure 8b. The white dots in Figure 8,b represent the weights, and the more white dots there are in each column, the more weight is added. Because the width between traffic lines is about 3 m, higher weights are applied to the point 1.5 m away from the left line, and lower weights are applied thereafter. The same method is applied to the right line. Moreover, if the weight for the left line and the one for right line are applied to the same column, the weight value is decided by superposition. Due to this weighting method, even if one line is hidden and not recognized, the highest weights are given to the center of the lane. In addition, even when the lane width is not constant, the highest weights can be given to the center of the lane, as shown in Figure 8c. The weighted lane field is formed by applying the above process to every row. The weighted lane field shown in Figure 8d is the result of the application of the above process to the one shown in Figure 7.

### 3.2.2. LiDAR Data

From the LiDAR data, 571 distance data points from $-45°$ to $225°$ based on the location of LiDAR data was obtained. After transforming these data to point data with ordinary coordinates, the points were clustered by classifying them as a single object. Based on the clustering process, the circle with the minimum size that fit the clustered points was determined as the object. In this process, the distance between neighboring points was less than 10 cm. Thus, the object data included the location and size of objects. In this process, the average filtering method was also applied to remove the spiking noise.

To prevent obstacle collision in advance, the field formed by LiDAR was divided into dangerous and safe areas. The fitted circles, which represent objects, were expressed by radius R and the center locations ($cenX$, $cenY$). As shown in Figure 9a, these circles represent the detected objects, and can be also represented with the following parameters: $tD$ is the distance between LiDAR position ($Lx$, $Ly$) and the tangent points of the object; $cenDist$ is the distance from ($Lx$, $Ly$) to the center of the object; $\angle l$ and $\angle s$ represent the angle to the center of the object and the half angle between two tangents, respectively. Moreover, the back area of the obstacles, which was not detected by LiDAR, was defined as the unknown area, being represented as form of a four-point polygon. Equation (1) expresses each point $\{(x1, y1), (x2, y2), (x3, y3), (x4, y4)\}$ forming the polygon.

$$
\begin{aligned}
x_1 &= tD * \cos(l - s) + L_x, \ y_1 = L_y - tD * \sin(l - s) \\
x_2 &= tD * \cos(l + s) + L_x, \ y_2 = L_y - tD * \sin(l - s) \\
x_3 &= (x_2 - L_x) * \tfrac{L_y - topY}{L_y - y2} + L_x, \ y_3 = topY \\
x_4 &= (x_1 - L_x) * \tfrac{L_y - topY}{L_y - y1} + L_x, \ y_4 = topY
\end{aligned}
\tag{1}
$$

As shown in Figure 9b, the LiDAR field was divided into an object area (red), unknown area (yellow), safe area (black). The red area represents obstacles and the yellow area is undetected, defined as the dangerous area. Before moving on to the next step, the object field formed as above is changed to a binary weighted field, as shown in Figure 10a, which shows the dangerous area in black and also shows the other area in gray.

However, rather than being represented by two parted areas, it is necessary to define which points are safer than other points in the safe area. Equation (2) expands the dangerous area by a constant width with the $M2$ value shown in Figure 10b.

$$
\begin{aligned}
&If, \ 0 \in \{M_1[x + i, y + i] * M_2[i, j] \big| i, j \in [0 : 9]\} \\
&Then, \ M_1[(x : x + 9), (y : y + 9)] = zeros_{[10 \times 10]}
\end{aligned}
\tag{2}
$$

M1 is a matrix field measuring 300 pixels × 300 pixels (6 m × 6 m), which is the sample field for LiDAR, as shown in Figure 10a. $M2$ is a window measuring 10 pixels × 10 pixels (0.2 m × 0.2 m), as shown in Figure 10b. As a result of Equation (2), the weighted object field is formed, as shown in Figure 10c.

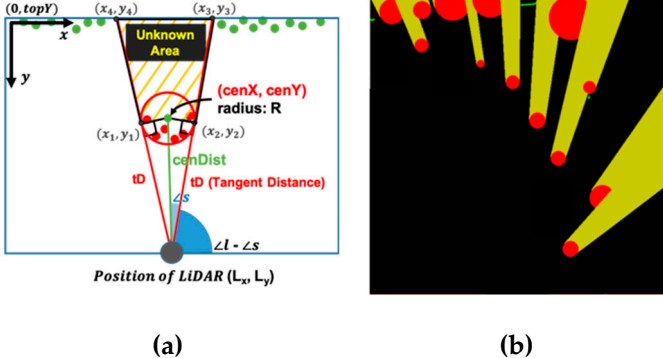

**Figure 9.** Area distinction of the obstacle field: (**a**) LiDAR field diagram; (**b**) sample of the object field.

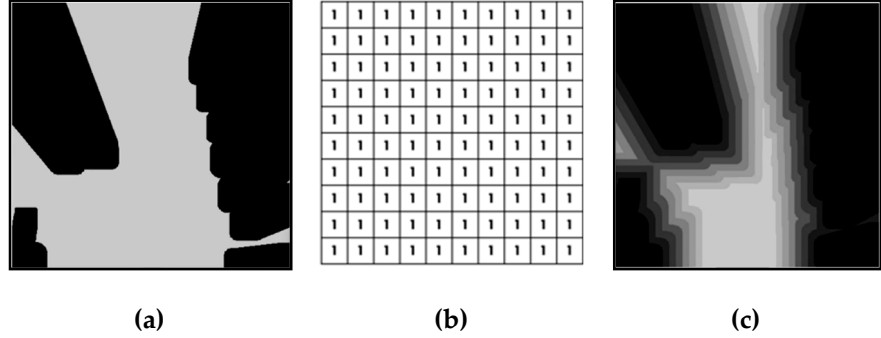

**Figure 10.** Method to form the object field: (**a**) binary object field; (**b**) window for offset; (**c**) weighted object field.

### 3.2.3. GPS Data

When traveling to a destination, the vehicle should keep the route that can be obtained from the map. The vehicle requires the relative route when traveling in order to go the right way, but the route data of the map is expressed as absolute coordinates $X_A O Y_A$. For this reason, Equations (3) and (4) are utilized. With the values from the GPS data, such as *My position O′* (which is the absolute location of the vehicle) and heading angle $\theta$, in Equation (3) the coordinate from $X_A O Y_A$ is changed to the relative coordinate $X_R O' Y_R$. Finally, in Equation (4), to synchronize lane and object fields, waypoints are transformed into the coordinate $X'_R O'' Y'_R$.

$$\begin{pmatrix} x_R \\ y_R \end{pmatrix} = \begin{pmatrix} \cos\theta & -\sin\theta \\ \sin\theta & \cos\theta \end{pmatrix} \begin{pmatrix} 1 & 0 \\ 0 & -1 \end{pmatrix} \begin{pmatrix} x_A - O'_x \\ y_A - O'_y \end{pmatrix} \tag{3}$$

$$\begin{pmatrix} x_R' \\ y_R' \end{pmatrix} = \begin{pmatrix} x_R - 3m \\ y_R - 6m \end{pmatrix} \tag{4}$$

In forming the weighted route field shown in Figure 11b, the waypoints and their surroundings are weighted. Based on the location of each waypoint, the surrounding area of the point is divided into circles of four different radii. The smaller the circle, the higher the weight that is given, and the outer area of the largest circle is not weighted. Additionally, for regions that overlap with other waypoints, the higher weight value out of the two is applied. Thus, when regions are closer to the waypoints, their weights are higher. This ensures that when an object is present on the path it is not completely off, while it is off the path when traveling.

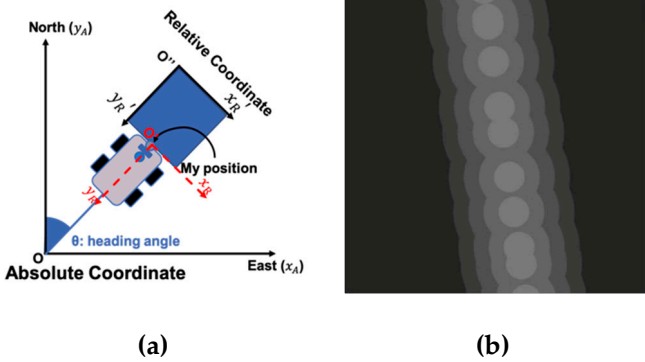

**(a)**          **(b)**

**Figure 11.** Method to process GPS data: (**a**) coordinate transformation of route data; (**b**) weighted route field.

### 3.2.4. SWIF Algorithm

By summing the three types of weighed fields, SWIF is formed for the situation shown in Figure 12a,b (shown in Figure 12c,d respectively). In SWIF, each coordinate has a weight ranging from 0 to 255, indicating the degree of safety.

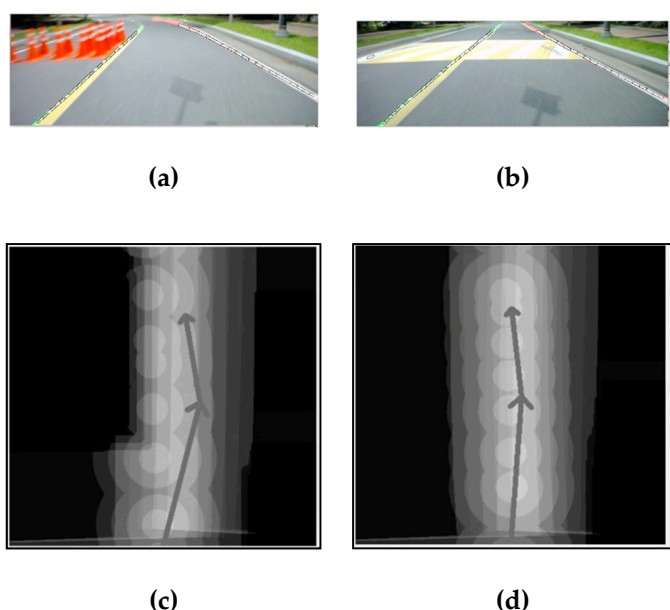

**(a)**          **(b)**

**(c)**          **(d)**

**Figure 12.** Driving situations and their SWIF values: (**a**) road image including obstacles; (**b**) road image including a speed bump; (**c**) result of SWIF algorithm in situation of (**a**); (**d**) result of SWIF algorithm in situation of (**b**).

### 3.3. Proposed Motion Planning and Maneuvering Control

Therefore, the SWIF algorithm is able to note whether each point is safer in the formed field. In order to judge which area (rather than which point) in the field is safer for maneuvering. The desired steering angle and speed of the vehicle are obtained by the cost field for motion planning. Consequently, these two variables are used for the reference input values in the closed-loop vehicle maneuvering controller, which is implemented with PID plus an integral anti-windup scheme to follow the safe area obtained by the SWIF algorithm.

### 3.3.1. Vehicle Speed and Steering Angle Decisions

The cost field, as shown in Figure 13a, has four parameters, *d*, *width*, *θ1*, and *θ2*. Here, *d* is the half height of the cost field as a scanning range, and *width* means the width of the vehicle. These two parameters are constant. On the other hand, *θ1* and *θ2* are the variables that vary in the cost field. Here, *θ1* is the candidate of the desired steering angle and *θ2* is the potential angle for checking whether the next area of *θ1* is safer or not. Thus, each cost field contains different *θ1* and *θ2* values.

The cost field is expressed as M3, and the sample of SWIF is shown as M4 in Equation (5) and Figure 13. The size of M3 and M4 is the same as 300 pixels × 300 pixels (or 6 m × 6 m). As shown in Equation (5), M3 and M4 are operated as a Hadamard product, and then the theta weighting field score is derived as a scalar value by summing all of the elements in the operated matrix. Moreover, by varying the thetas, different scores are derived.

$$Score\ of\ n_{th}\ theta\ weighting\ fieldscore : S_n$$
$$S_{max} = \max\{S_n | S_n = \sum_{i=0}^{i=299} \sum_{i=0}^{j=299} (M_3[i,j] * M_4[i,j])\} \tag{5}$$

Finally, as shown in Figure 13c, the *θ1* and *θ2* are determined as the values of the cost field from which the maximum score is derived. Thus, *θ1* becomes the desired steering setting.

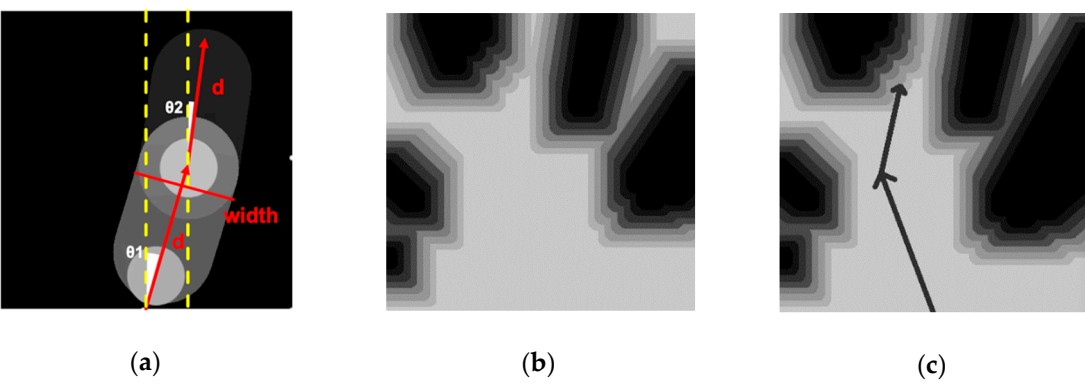

|　(a)　|　(b)　|　(c)　|

**Figure 13.** Motion planning method: (**a**) sample of the cost field; (**b**) sample of SWIF; (**c**) result represented in SWIF.

### 3.3.2. Maneuvering Control Algorithm

Using the SWIF algorithm with SSCNN, the desired steering angle and speed values are derived with the sampling rate of 20 Hz. Since a sudden change of the vehicle steering angle is obtained with the instantaneous sensor recognition, vehicle maneuver with a sudden large change in steering angle or speed may occur. To prevent this situation in advance and to allow flexible driving, outputs (i.e., steering angle and vehicle speed) of the PID were moderated by the integral anti-windup scheme, as shown in Figure 14.

As shown Figure 14a, reference input variables *v* and *θ* are the desired speed and steering angle values, respectively, as in the results of the algorithm described above, and variables *v_out* and *θ_out* are the output values calculated by the proposed algorithm. In order to track the desired values, a PID controller for vehicle speed was designed to provide a fast response to change of desired speed by reducing the selection time. However, PID control parameters for the steering angle were finely tuned, not only to provide a relatively slow response, but also to avoid a rocking motion when driving. Moreover, the integral anti-windup scheme in the PID control loop, as shown in Figure 14b, was utilized to prevent dangerous situations, such as sudden stopping or sudden acceleration.

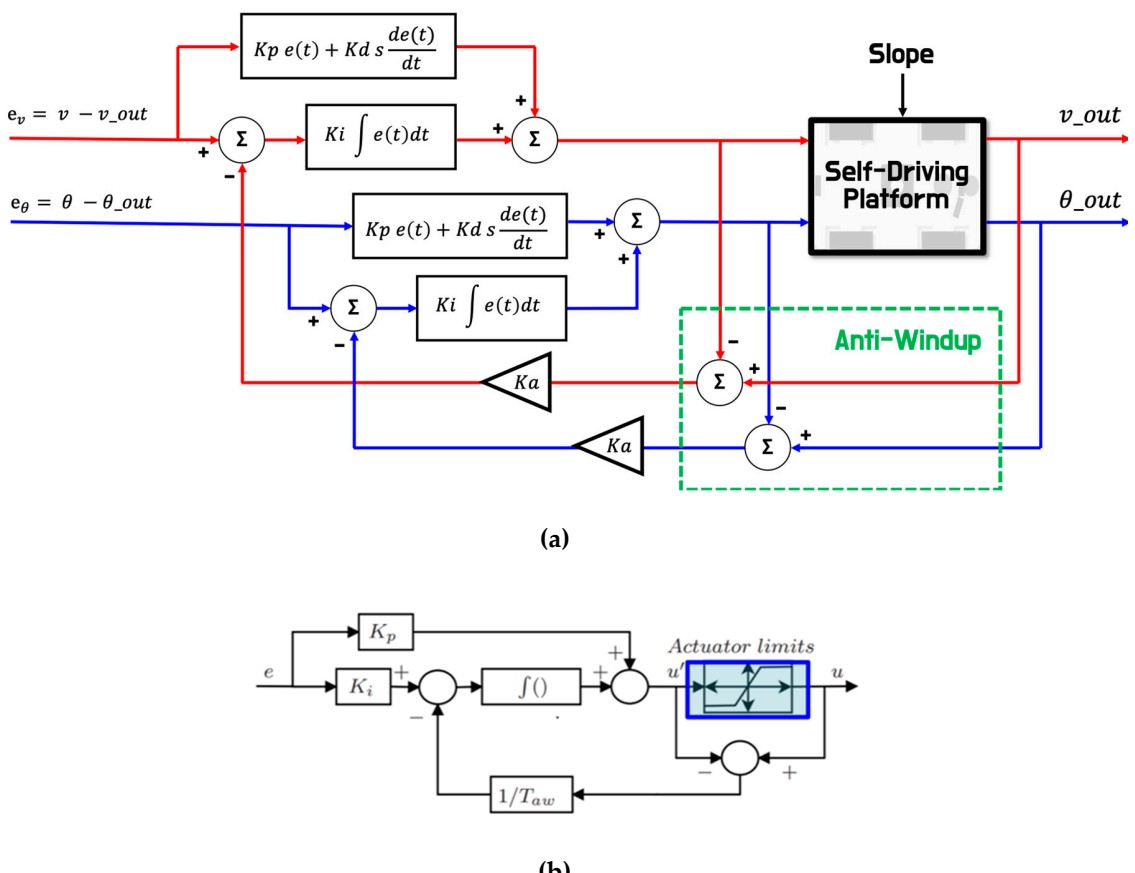

**Figure 14.** Vehicle speed and steering angle control diagram: (**a**) used overall control block diagram; (**b**) specific parameters of integral anti-windup scheme.

## 4. Experimental Results

To verify the performance of the proposed algorithm, the algorithm was tested on diverse roads using both K-City and Daegu Gyeongbuk Institute of Science & Technology (DGIST) campus including straight roads, curved roads, crosswalks, bumps, road markings, school zones, and bus lanes. For the obstacle conditions, the proposed algorithm was also tested for situations where a dynamic obstacle suddenly enters the lane or a large obstacle blocks a large part of the road.

### 4.1. Lane Recognition with Sparse Spatial CNN

To recognize the traffic lane in real time, the vision deep learning result was also included. Tables 3 and 4 show the accuracy comparison between the newly proposed SSCNN method and the existing SCNN method. As shown in Table 3, when Tusimple or KURD is used as the dataset, the accuracy is similar, but the proposed SSCNN method has 2.7 times faster processing speed than the original one. On the other hand, as shown in Table 4, in using CULane the accuracy of the proposed SSCNN method is on average 5.4% lower than the original one for all classes except crossroads.

**Table 3.** Performance comparison.

| Tusimple/KURD | SCNN [6] | SSCNN |
|---------------|----------|-------|
| Accuracy      | 94.62%   | 94.56% |
| Time (s)      | 0.124    | 0.0459 |
| Speed (fps)   | 8.063    | 21.796 |

Units of accuracy: F1 measure.

**Table 4.** Accuracy comparison.

| CULane | SCNN [6] | SSCNN |
|---|---|---|
| Normal | 90.6 | 83.0 |
| Crowded | 69.7 | 64.1 |
| Night | 66.1 | 62.1 |
| No line | 43.4 | 39.2 |
| Shadow | 66.9 | 57.9 |
| Arrow | 84.1 | 78.5 |
| Dazzle light | 58.5 | 56.5 |
| Curve | 64.4 | 54.9 |
| Crossroad | 1990 | 2759 |
| Total | 71.6 | 66.2 |

Units: F1 measure. (Crossroad: FP measure.)

*4.2. Test Scenario 1: Pedestrian in the Lane*

In the first obstacle test, a scenario in which a pedestrian suddenly entered into the lane was accomplished, as shown in Figure 15. In Figure 16, the blue line represents the planned waypoints at the center of the lane and the red line is the real path where the vehicle was maneuvered. Moreover, as shown in Figure 16a, just before and after avoiding the pedestrian, the vehicle traveled along the center of the lane. As a result, as shown in Figure 16b, when a pedestrian invaded the lane, the vehicle traveled with an average deflection of 0.70 m from the center without lane departure and recovered straight after avoiding the pedestrian.

As shown in Figure 17, the graph represents the comparison between the desired steering angle (red line) and the actual steering angle (blue line), and shows that the vehicle traveled without overshooting the steering angle. Specifically, at the 7 s timepoint the vehicle avoided a pedestrian, and at 13 s returned to normal autonomous driving.

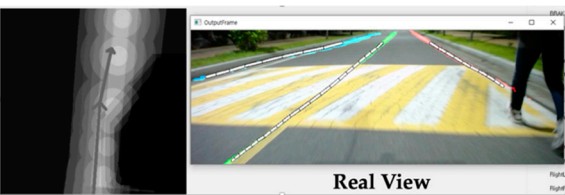

**Figure 15.** Real-time results for test scenario 1.

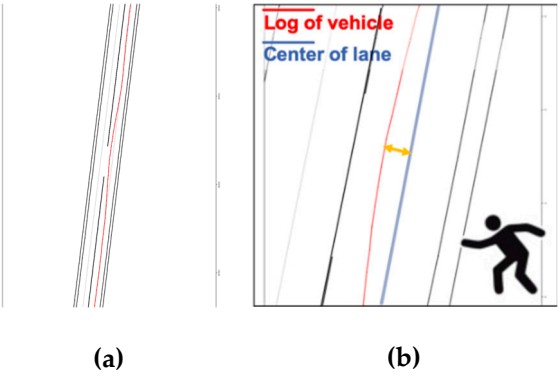

(a)                    (b)

**Figure 16.** Comparison between maneuvered path and waypoints in scenario 1: (**a**) tracked path: (**b**) magnified path while avoiding pedestrian.

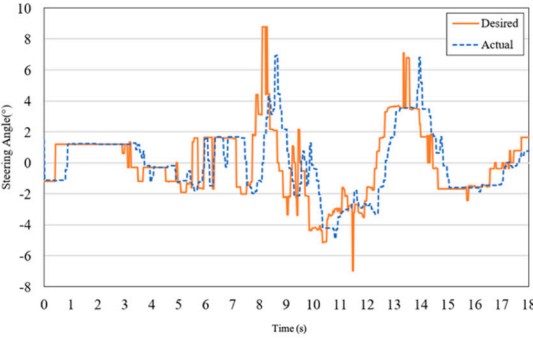

**Figure 17.** Desired steering angle and controlled steering angle in driving in test scenario 1.

### 4.3. Test Scenario 2: Construction Site on the Road

As shown in Figure 18, for the second obstacle test, a scenario in which a construction site blocked part of road was applied to check the performance of the proposed algorithm. The blue and red lines in Figure 19 have same meaning as for the previous test in Section 4.2. As shown in Figure 19a, the vehicle was maneuvered in advance to avoid a construction site. Because the vehicle avoidance took a longer time than in the previous scenario, the variation of the steering angle was smaller, as shown in Figure 20. As shown in Figure 19b, the deflection is 0.82 m, which is larger than for scenario 1. Likewise, there is no lane or route departure of vehicle overshoot in steering.

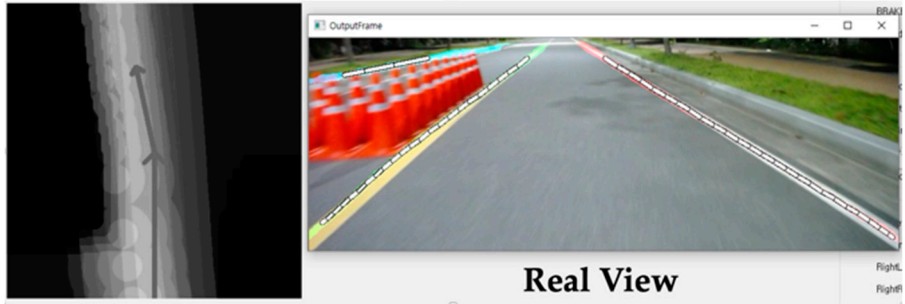

**Figure 18.** Real-time results for test scenario 2.

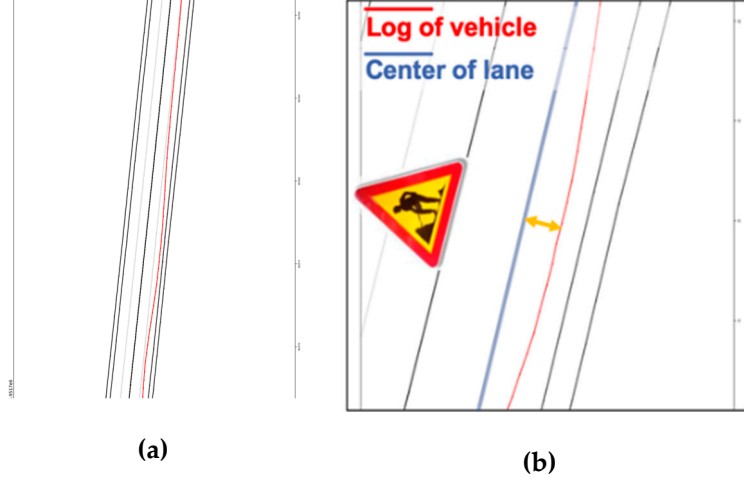

(a)                          (b)

**Figure 19.** Comparison between maneuvering path and waypoints in scenario 2: (**a**) tracked path; (**b**) magnified path while avoiding construction site.

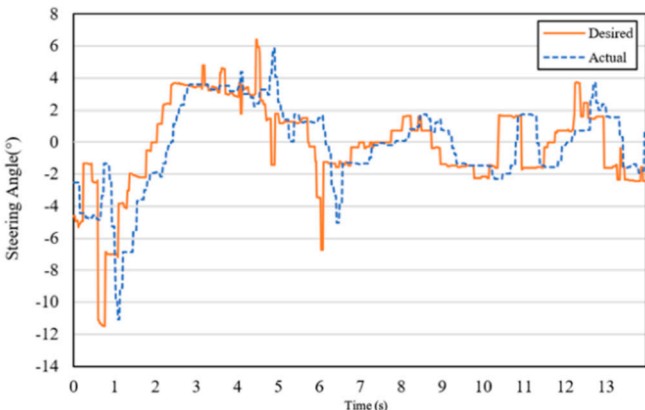

**Figure 20.** Desired steering angle and controlled steering angle for driving in scenario 2.

*4.4. Performance of Proposed Algorithm in International College Creative Car Competition*

The reliability of the proposed autonomous driving algorithm was verified by being used in real urban road environments (i.e., K-City in Korea). As shown in Figure 21, different road environments were implemented, such as a school zone, crossroad, crosswalks, bicycle and bus lanes, and stationary vehicles. The performance was the same as described above, successfully proving that the proposed autonomous driving algorithm was significantly convincing in the presence of diverse road conditions.

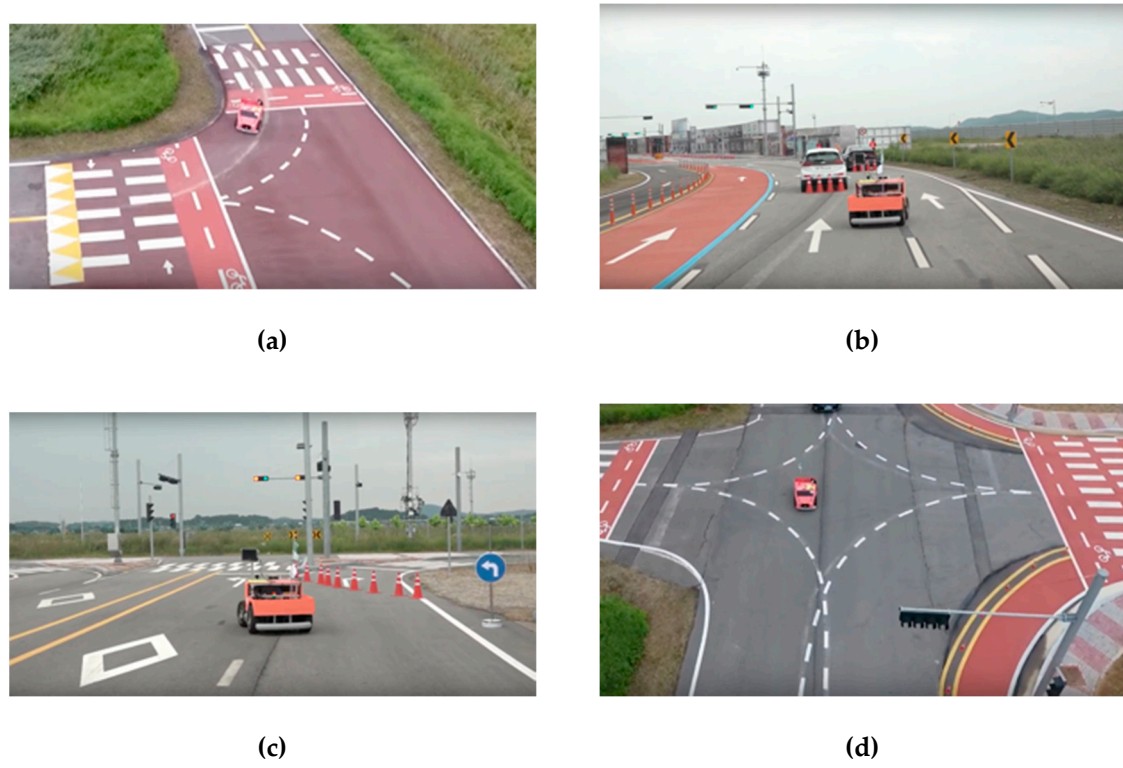

**Figure 21.** Tests in K-City: (**a**) driving in school zone; (**b**) static obstacle; (**c**) lane change; (**d**) intersection driving

## 5. Discussion and Remarks

By changing the existing network structure of SCNN to be sparser, the accuracy of SSCNN becomes lower by 5.4% on average than the original SCNN. However, the average recognition rate is 66.2% in various urban environments. Additionally, when it is applied to SWIF, the result is complemented by using the averaging filter that enables the derived value to be skipped when a large deviation from the

previous results is derived. Moreover, since SSCNN is capable of producing values 2.7 times faster than the original one, it is more suitable for real-time systems and is able to compensate for slightly lower accuracy.

Both the SWIF and control algorithms are tested in two urban scenarios: (a) a person suddenly appearing on the road (b) and a construction site covering part of the road. As a result, even when large objects blocked part of the road or obstacles appeared at close ranges, it was possible to observe the slow speed and rotation of the steering angle, which helped avoid a secondary dangerous situation due to oversteering. Moreover, considering the presence of objects outside the lane, it was also able to maneuver without lane or route deviation by keeping a safe distance (i.e., about 50 cm) from the objects.

## 6. Conclusions

This paper proposed two novel algorithms: the first algorithm is the vision deep learning method, which is named the sparse spatial convolutional neural network (SSCNN); and the second one is the sensor integration method, which is called the sensor-weighted integration field (SWIF). Due to the proposed SSCNN, for lane recognition on urban roads, the test vehicle was able to recognize the adjacent traffic lanes in any direction on urban roads, since lane data was learned by considering both normal and abnormal driving directions at the same time. Moreover, it also worked well because of the newly proposed network model, which is sparser than the existing one. Thus, when learned with the same Tusimple and CULane datasets, the accuracy of SSCNN was lower but the computational speed was dramatically increased by 2.7 times, which significantly contributed to real-time system performance for the autonomous vehicle. For this reason, it is noted that SSCNN seems to be adoptable. In the future, a slight reduction in accuracy will be corrected in the next step, such as integrating sensors or applying filters. Based on the detected lanes and the previously described SSCNN, the SWIF algorithm was also proposed by forming a weighted field, utilizing both the obstacle data from LiDAR data and waypoints from GPS data. This system is efficient, indicating which areas are safe from dangerous factors, with a resolution of 2 cm per pixel and a processing speed of 18–26 frames per second. Additionally, SWIF can simplify the data integration and can be expanded easily without requiring complicated mathematical calculations. Using the motion planning method in SWIF, this algorithm does not always judge the center area between feature points (e.g., the lanes, routes, and obstacles) as a safe path. The safe path is decided in two steps ($\theta 1$, $\theta 2$), so that the vehicle can detect the safest direction and area and keep a safe distance from dangerous factors on diverse urban roads and with minimum change of steering angle. Consequently, as the SWIF algorithm is based on SSCNN, to prevent a dangerous situation in advance, the vehicle can recognize situations both inside the lane and outside the lane and then divert the travel direction from the center path for safer urban driving, rather than just following the center line between lanes and obstacles. Thus, the vehicle is able to travel in real-time with flexibility and without route or lane departure due to the sudden steering in the presence of diverse disturbances on urban roads. Moreover, in the future, efficient paths calculated from diverse information obtained from V2X communication will be also integrated with the proposed SWIF algorithm for better autonomous driving performance in urban environments.

**Author Contributions:** Conceptualization, M.O., B.C., I.B., G.C. and Y.L.; Data curation, M.O., B.C. and I.B.; Investigation, M.O., B.C., I.B., G.C. and Y.L.; Methodology, M.O.; Project administration, M.O.; Software, M.O., B.C. and I.B.; Supervision, G.C. and Y.L.; Validation, M.O., B.C. and I.B.; Visualization, M.O., B.C. and I.B.; Writing—original draft, M.O., B.C. and I.B.; Writing—review & editing, G.C. and Y.L. All authors have read and agreed to the published version of the manuscript.

**Acknowledgments:** The authors would like to thank the Undergraduate Group Research Project (UGRP) program of Daegu Gyeongbuk Institute of Science & Technology (DGIST) for the research funding support.

**Conflicts of Interest:** The authors declare no conflict of interest.

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
