# Peer review of "An Urban Autodriving Algorithm Based on a Sensor-Weighted Integration Field with Deep Learning"

_electronics, doi:10.3390/electronics9010158_

Round 1

Reviewer 1 Report

The Authors have presented a method to control autonomous vehicle based on data fusion from vision, LiDAR, and GPS sensors. The undertaken topic is interesting and worth to be investigated. However, the manuscript cannot be published in its current form due to its low quality.

An extended discussion of state of the art in the field of traffic lane detection and sensor fusion has to be included in the paper with appropriate references to recent publications.

In this manuscript the description of the proposed method is not clear and does not allow the interested researchers to implement the algorithms and replicate the experiments.

In Section  3.1.2 the proposed network model should be better described. The meaning of symbols C, W, and n is not defined. The concept of sparse spatial CNN is not sufficiently presented. What is the role of the neurons without connections in Fig. 5 c - e?.

In Section 3.2.1 the method of lane detection by Sparse SCNN  is not sufficiently presented. Authors should discuss in details the transformation of lane location data to top view and the computation of "lane field" (formulas and/or pseudo-codes are necessary). Similarly, the " weighting method " is not clearly presented. The drawings in Fig. 7 have to be clearly described in the text. For instance: What is the meaning of the white, yellow and red squares in Fig. 7  a - b.

Please provide more information regarding the noise reduction and clustering operations performed for LIDAR data in Sect 3.2.2. Similarly, the " binarization and reveral" have to be explained in details.

Does Fig. 9 c show the well-known distance transfer? (see Fabbri, R., Costa, L.D.F., Torelli, J.C. and Bruno, O.M., 2008. 2D Euclidean distance transform algorithms: A comparative survey. ACM Computing Surveys (CSUR), 40(1), p.2.)

In Sect. 3.2.3 an algorithm for calculating the weighted route field (as shown in Fig. 10 b) should be provided. The results shown in Fig. 10 b seems to be incorrect because the darker road sections between "waypoints" along the road are labeled as unsafe (dark color).

The Maneuvering Control Algorithm shown in Fig. 13 should be better described.

The discussion of experimental results and conclusions has to be extended.

Sect 3.2: "informs the safer area than normal maneuvering" - What do you mean by " normal maneuvering "?

The numbers in figure captions are not correct. The manuscript includes two figures with number "1".

The paper is hard to follow due to the paper is hard to follow, due to poor English and an unusual number of awkward sentences. Few examples are given below.

Page 2:

"A few years ago, most conventional algorithms manually checked low level features, such as color patterns and straight lines [7, 8]." Please consider revising this sentence. Color patterns and straight lines can be extracted automatically by conventional algorithms.

"This method enables to expect obscured lane by extending." - this sentence needs revision. The idea has to be clearly stated and better discussed.

"SCNN can be improved by more sparsely approach" - the meaning of "more sparsely approach" should be clearly explained.

"To Determine the motion of autonomous vehicles, information about road condition is integrated." - please explain what means " information about road condition ". Use lowercase "d" for the second word in this sentence.

" Moreover, road coverage by sensors is shown in Fig. 2." - The reference to figure is incorrect. Fig. 2 shows an overview of the proposed algorithm.

Page 3:

" In the case of Caltech Lanes Dataset (Aly 2008), Tusimple Benchmark Dataset (Tusimple 2017), and CULane Dataset (Pan 2018) datasets" - please add correct references for the datasets.

Author Response

Dear Reviewer,

We made the following modifications.

The Authors have presented a method to control autonomous vehicle based on data fusion from vision, LiDAR, and GPS sensors. The undertaken topic is interesting and worth to be investigated. However, the manuscript cannot be published in its current form due to its low quality.

An extended discussion of state of the art in the field of traffic lane detection and sensor fusion has to be included in the paper with appropriate references to recent publications.

We revised manuscript by adding the recent researches in introduction section, and discuss them.

In this manuscript the description of the proposed method is not clear and does not allow the interested researchers to implement the algorithms and replicate the experiments.

We revised by adding more details of SSCNN and SWIF, such as method of sensor data pre-processing, and explanation of figures (Fig 5, 7, 8, 9, 11, 13, 14)

In Section 3.1.2 the proposed network model should be better described. The meaning of symbols C, W, and n is not defined. The concept of sparse spatial CNN is not sufficiently presented. What is the role of the neurons without connections in Fig. 5 c - e?.

We revised manuscript by adding the details of SSCNN; Adding the meaning of C, H, W and n about the structure of the proposed network model. Moreover, the explanation of Figure 6.c-e was added for better understanding.

In Section 3.2.1 the method of lane detection by Sparse SCNN is not sufficiently presented. Authors should discuss in details the transformation of lane location data to top view and the computation of "lane field" (formulas and/or pseudo-codes are necessary). Similarly, the " weighting method " is not clearly presented. The drawings in Fig. 7 have to be clearly described in the text. For instance: What is the meaning of the white, yellow and red squares in Fig. 7 a - b.

Please provide more information regarding the noise reduction and clustering operations performed for LIDAR data in Sect 3.2.2. Similarly, the " binarization and reveral" have to be explained in details.

We revised manuscript by adding the details of pre-processing method, and algorithm flow for better understanding about weighted lane field and weighted object field.

Does Fig. 9 c show the well-known distance transfer? (see Fabbri, R., Costa, L.D.F., Torelli, J.C. and Bruno, O.M., 2008. 2D Euclidean distance transform algorithms: A comparative survey. ACM Computing Surveys (CSUR), 40(1), p.2.)

I know that the “Euclidean distance transform” is used in A-star. But we didn’t use the A-star algorithm. So, we calculate the distance with classical meth (Pythagorean law).

In Sect. 3.2.3 an algorithm for calculating the weighted route field (as shown in Fig. 10 b) should be provided. The results shown in Fig. 10 b seems to be incorrect because the darker road sections between "waypoints" along the road are labeled as unsafe (dark color).

Since we use the waypoints as a “point”. If we use the route as a line, we can avoid the situation we concerned. But GPS data often splatters and then gives confusion about the order of points that have a bad effect on the weighted field. This is why we used the waypoints as a “point”.

The Maneuvering Control Algorithm shown in Fig. 13 should be better described.

We revised manuscript by updating with the additional paragraph in Section 3.3.2 for explaining the details of variables in the Fig. 14.

The discussion of experimental results and conclusions has to be extended.

We added newly “Discussion and remarks section” for discussion of result and evaluation and comparison of algorithm.

Sect 3.2: "informs the safer area than normal maneuvering" - What do you mean by " normal maneuvering "?

We revised manuscript by changing the word.

The numbers in figure captions are not correct. The manuscript includes two figures with number "1".

We corrected the figure numbers.

The paper is hard to follow due to the paper is hard to follow, due to poor English and an unusual number of awkward sentences. Few examples are given below.

Page 2:

"A few years ago, most conventional algorithms manually checked low level features, such as color patterns and straight lines [7, 8]." Please consider revising this sentence. Color patterns and straight lines can be extracted automatically by conventional algorithms.

"This method enables to expect obscured lane by extending." - this sentence needs revision. The idea has to be clearly stated and better discussed.

"SCNN can be improved by more sparsely approach" - the meaning of "more sparsely approach" should be clearly explained.

"To Determine the motion of autonomous vehicles, information about road condition is integrated." - please explain what means " information about road condition ". Use lowercase "d" for the second word in this sentence.

" Moreover, road coverage by sensors is shown in Fig. 2." - The reference to figure is incorrect. Fig. 2 shows an overview of the proposed algorithm.

Page 3:

" In the case of Caltech Lanes Dataset (Aly 2008), Tusimple Benchmark Dataset (Tusimple 2017), and CULane Dataset (Pan 2018) datasets" - please add correct references for the datasets.

Not only these examples, we also revised the overall English expressions by several proof reading.

Reviewer 2 Report

In this paper, an urban auto-driving algorithm is proposed. Based on vision deep learning, sparse spatio CNN (SSCNN) and the proposed sensor weighted integration field (SWIF), the vision sensor, LiDAR, and GPS sensors are used to decide critical motion. The research topic is interesting, but the technical details are so limited. For example, 3.1.2, only less than 10 lines for the proposed SCNN. In p.6, the description above Figure 8 are conflict. The figure number are wrong. In addition, the authors should compare the proposed with the state-of-the-art methods to make the paper more convincing.

Author Response

Dear Reviewer,

We made the following modifications.

"For example, 3.1.2, only less than 10 lines for the proposed SCNN."

We added the details of SSCNN; Adding the meaning of C, H, W and n about the structure of the proposed network model. Moreover, the explanation of Figure 6.c-e was added in the manuscript.

"In p.6, the description above Figure 8 are conflict. The figure number are wrong."

We corrected the Figure numbers. Additionally, we added what Figure 8. a and b mean, and function of them. The details about weighted lane field and weighted object field were added, such as the pre-processing of them. (with more explanation or Pseudocode)

"In addition, the authors should compare the proposed with the state-of-the-art methods to make the paper more convincing."

We additionally refer some recent researches about the lane detection and the sensor fusion algorithm in Section 1.

In addition, many English expressions have been corrected, and insufficient algorithmic descriptions have been corrected and supplemented in the text.

Reviewer 3 Report

This paper studies  adaptive driving on the urban environment. Although the paper has some merits, I have few general concerns that need to be addressed carefully before I can recommend for publications:

  (1) Could you please explain the applicability of the proposed schemes in intersection scenario?   (2) The term “outstanding” in the abstract is not appropriate. Please remove it.    (3) The texts need improvement as typos and grammatical mistakes are there.   (4) "CNN" term is used  in the beginning without any explanation.     (5) Not many recent V2X papers are discussed and cited in the paper. We strongly suggest to include following recent papers while discussing the proposed technique's applicability for DSRC and LTE V2X network.   (i) Noor, "Broadcast Performance Analysis and Improvements of the LTE-V2V Autonomous Mode at Road Intersection," IEEE TVT, 2019.   (ii) Jianhua He 'Enhanced Collision Avoidance for Distributed LTE Vehicle to Vehicle Broadcast Communications', 2018   (iii) Noor, "A Survey on Resource Allocation in Vehicular Networks," arXiv, 2019.   (iv) Francisco et al 'Geolocation-Based Access for Vehicular Communications: Analysis and Optimization via Stochastic Geometry', 2018.     (6) Please rewrite the motivation of the proposed work clearly.   (7) Authors should improve the quality of Fig. 13.   (8) Please state future directions.

Author Response

Dear Reviewer,

We made the following modifications.

Could you please explain the applicability of the proposed schemes in intersection scenario? We checked whether this algorithm works well in intersection scenario with the route data from GPS and the object data from LiDAR, because there are so many lanes confused to be recognized in intersection. We confirmed that the algorithm was applicable in the urban environment (K-city).

The term “outstanding” in the abstract is not appropriate. Please remove it.    We remove that expression, not only in the abstract, but also in the conclusion section.

The texts need improvement as typos and grammatical mistakes are there.   We revised the overall paper.

"CNN" term is used in the beginning without any explanation.     We added the meaning of CNN in the abstract.

Not many recent V2X papers are discussed and cited in the paper. We strongly suggest to include following recent papers while discussing the proposed technique's applicability for DSRC and LTE V2X network.   (i) Noor, "Broadcast Performance Analysis and Improvements of the LTE-V2V Autonomous Mode at Road Intersection," IEEE TVT, 2019.   (ii) Jianhua He 'Enhanced Collision Avoidance for Distributed LTE Vehicle to Vehicle Broadcast Communications', 2018   (iii) Noor, "A Survey on Resource Allocation in Vehicular Networks," arXiv, 2019.   (iv) Francisco et al 'Geolocation-Based Access for Vehicular Communications: Analysis and Optimization via Stochastic Geometry', 2018.     Thus, we introduced V2X technology and discussed about how to apply the V2X tech and our proposed algorithm together for urban driving system, as a future direction.

Please rewrite the motivation of the proposed work clearly.   We revised the introduction of our proposed algorithm with more explanation, such as functions and purpose of SSCNN and SWIF, and Synergy of them in this algorithm.

Authors should improve the quality of Fig. 13.   We revised that the Figures become bigger for clear viewing. This figure notes that there is no blur situation - the many stepped lines in them are the values that we have calculated and connected.

Please state future directions. We added the future directions of our proposed algorithm in “Conclusion” section. In this part, we deal with how to apply our proposed algorithm for urban driving system with other recent researches.

In addition, many English expressions have been corrected, and insufficient algorithmic descriptions have been corrected and supplemented in the text.

Round 2

Reviewer 1 Report

The Authors have slightly improved the manuscript. However, further revision is necessary to provide comprehensive and clear information about details of the proposed approach.

Section 3.1.2 should be rewritten. The proposed network model should be better described. The concept of sparse spatial CNN is not clearly presented. The difference between MRF/CRK, spatial CNN and the proposed sparse spatial CNN has to be explained in details by extending the discussion of the examples presented in Fig. 6. What are the inputs and outputs of the considered neural network? What information is conveyed between "slices"? It is not clear if the Authors have just modified structure of the neural network or the training algorithm has been also modified. What are the values of C, H, W and n for the neural networks shown in Fig. 6? Where is start and end of the "message passing path" in Fig. 6? Are the "messages" processed by neurons?

"It is confirmed that the algorithm takes a lot of time for the sequential progress of the overall network execution time, and also proposes a message passing method by dividing the constant value n into n / H slices." - this sentence is completely unclear. Which algorithm? Where it is confirmed?

The last sentence in Sect. 3.1.2 is not logically connected with the previous part of that Section: "In real-world application, argmax is used to get the maximum value of pixel and also applied Gaussian curve to smoothen the output values." - What is the connection between pixels and the presented tensor-based model of neural network? Please note that argmax function does not return the maximum value, thus the statement is confusing. How the parameters of "Gaussian curve" were set.

In Section 3.2.1 the method of lane detection is not sufficiently presented. How the lanes are detected, i.e., how the blue, red and green lines in Fig. 7 were determined? The Authors have stated that "As shown in Fig. 7, the result of SSCNN means the location of the four adjacent lanes in the road image". However, Fig.7 shows an image of road with two traffic lanes.

Authors should discuss in details the transformation of lane location data to top view (formulas and/or pseudo-codes are necessary).

The pseudo-code in Tab. 2 is unsatisfactory because the operations are not precisely defined. For instance : "For each row of the lane field Based on the column component of the left lane, the weights as shown in Fig. 8 (a) are assigned." is not correct because the row of the "lane field" has 300 pixels, while the grid shown in Fig. 8 has 20 elements per row. Mathematical formulas should be added to explain how the pixel values in Fig 8 c are calculated. The drawings in Fig. 8 have to be clearly described in the text. What is the meaning of the white, yellow and red squares in Fig. 7 a, b? Please add description of the axes.

In Sect. 3.2.3 an algorithm for calculating the weighted route field (as shown in Fig. 11 b) should be provided. Mathematical formulas should be added to explain how the pixel values in Fig 11 b are calculated.

Author Response

Dear Reviewer,

We modified our paper with following comments.

About 3.1

We rewrite and revised the manuscript of all parts in 3.1 by adding the specific expressions. For example, “why SSCNN is needed”, “what different things between SCNN and SSCNN are”, “what input and output of SSCNN are”, and so on. Specially, in Sect 3.1.2, many details were added, and the description sentences are increased over 25 lines.

============================================

About 3.2

In Section 3.2.1 the method of lane detection is not sufficiently presented. How the lanes are detected, i.e., how the blue, red and green lines in Fig. 7 were determined? The Authors have stated that "As shown in Fig. 7, the result of SSCNN means the location of the four adjacent lanes in the road image". However, Fig.7 shows an image of road with two traffic lanes.

The detected four lines are the results of SSCNN, a vision deep learning. “left-left: blue”, “left: green”, “right: red”, and “right-right: yellow”. These are deep learning results, deriving in four adjacent lanes above the lane. (Same expression of SCNN) And also, if the right-right line is missed, like Fig. 7, the result is shown as "none". We all revised and stated these in the manuscript.

Authors should discuss in details the transformation of lane location data to top view (formulas and/or pseudo-codes are necessary).

The used transformation of lane location is commonly used in other researches. Since we didn't develop it ourselves, it did not need to be mentioned. Instead of adding the code, we mentioned what the function is used. (“warping function of OpenCV”)

The pseudo-code in Tab. 2 is unsatisfactory because the operations are not precisely defined. For instance : "For each row of the lane field Based on the column component of the left lane, the weights as shown in Fig. 8 (a) are assigned." is not correct because the row of the "lane field" has 300 pixels, while the grid shown in Fig. 8 has 20 elements per row. Mathematical formulas should be added to explain how the pixel values in Fig 8 c are calculated. The drawings in Fig. 8 have to be clearly described in the text. What is the meaning of the white, yellow and red squares in Fig. 7 a, b? Please add description of the axes.

The Fig 8. (a), (b) are the schematic diagram of applied weights. The width of Fig 8(a, b) does not mean the width of lane field. In this regard, we added a specific description to the text and the Fig 8 was changed a lot for the detail expression. Moreover, the details of weighting method are added.

In Sect. 3.2.3 an algorithm for calculating the weighted route field (as shown in Fig. 11 b) should be provided. Mathematical formulas should be added to explain how the pixel values in Fig 11 b are calculated.

We added the specific way in text.

Reviewer 2 Report

In this paper, most of my raised issues are responded. However, for a possible publication, the authors can consider the following suggestions:

3.2, “Sensor Weighted Integration Field (SWIF)” => “Proposed Sensor Weight Integration Field (SWIF)” 3.3, “Motion Planning and Maneuvering Control” => Proposed Motion Planning and Maneuvering Control” Table 3, the title in the table “SCNN” can add a reference number “SCNN [?] ”

Author Response

Dear Reviewer,

We modified our paper with the following comments.

3.2, “Sensor Weighted Integration Field (SWIF)” => “Proposed Sensor Weight Integration Field (SWIF)” 3.3, “Motion Planning and Maneuvering Control” => Proposed Motion Planning and Maneuvering Control” Table 3, the title in the table “SCNN” can add a reference number “SCNN [?] ” 

I changed the expression and modified it as follows. “3.2 Sensor Weighted Integration Field (SWIF) -> 3.2 Proposed Sensor Integration Algorithm: Sensor Weighted Integration Field (SWIF)” for be similar to the expression in 3.1 Moreover, we added the reference number “[6]” for SCNN in Table 3, 4.

Reviewer 3 Report

We are satisfied with the revised version.

Author Response

Dear Reviewer,

Thank you for reviewing our paper.

Round 3

Reviewer 1 Report

The manuscript has been improved and now can published in Electronics.